# Anti-Oxidized Self-Assembly of Multilayered F-Mene/MXene/TPU Composite with Improved Environmental Stability and Pressure Sensing Performances

**DOI:** 10.3390/polym16101337

**Published:** 2024-05-09

**Authors:** Zhong Zheng, Qian Yang, Shuyi Song, Yifan Pan, Huan Xue, Jing Li

**Affiliations:** Hubei Key Laboratory of Modern Manufacturing Quantity Engineering, School of Mechanical Engineering, Hubei University of Technology, Wuhan 430068, China; zhengzh@hbut.edu.cn (Z.Z.); rain20220204@163.com (Q.Y.); songshuyi1016@163.com (S.S.); yuunagi0824@outlook.com (Y.P.); stonemechanics@163.com (H.X.)

**Keywords:** electrostatic self-assembly, functionalized-MXene, multilayer pressure sensor, inoxidizability

## Abstract

MXenes, as emerging 2D sensing materials for next-generation electronics, have attracted tremendous attention owing to their extraordinary electrical conductivity, mechanical strength, and flexibility. However, challenges remain due to the weak stability in the oxygen environment and nonnegligible aggregation of layered MXenes, which severely affect the durability and sensing performances of the corresponding MXene-based pressure sensors, respectively. Here, in this work, we propose an easy-to-fabricate self-assembly strategy to prepare multilayered MXene composite films, where the first layer MXene is hydrogen-bond self-assembled on the electrospun thermoplastic urethane (TPU) fibers surface and the anti-oxidized functionalized-MXene (f-MXene) is subsequently adhered on the MXene layer by spontaneous electrostatic attraction. Remarkably, the f-MXene surface is functionalized with silanization reagents to form a hydrophobic protective layer, thus preventing the oxidation of the MXene-based pressure sensor during service. Simultaneously, the electrostatic self-assembled MXene and f-MXene successfully avoid the invalid stacking of MXene, leading to an improved pressure sensitivity. Moreover, the adopted electrospinning method can facilitate cyclic self-assembly and the formation of a hierarchical micro-nano porous structure of the multilayered f-MXene/MXene/TPU (M-fM^2^T) composite. The gradient pores can generate changes in the conductive pathways within a wide loading range, broadening the pressure detection range of the as-proposed multilayered f-MXene/MXene/TPU piezoresistive sensor (M-fM^2^TPS). Experimentally, these novel features endow our M-fM^2^TPS with an outstanding maximum sensitivity of 40.31 kPa^−1^ and an extensive sensing range of up to 120 kPa. Additionally, our M-fM^2^TPS exhibits excellent anti-oxidized properties for environmental stability and mechanical reliability for long-term use, which shows only ~0.8% fractional resistance changes after being placed in a natural environment for over 30 days and provides a reproducible loading–unloading pressure measurement for more than 1000 cycles. As a proof of concept, the M-fM^2^TPS is deployed to monitor human movements and radial artery pulse. Our anti-oxidized self-assembly strategy of multilayered MXene is expected to guide the future investigation of MXene-based advanced sensors with commercial values.

## 1. Introduction

MXene is a new type of two-dimensional transition metal carbide/carbonitride. The classical formula is M_n+1_X_n_T_x_, where M represents the transition metal, X represents carbon or nitrogen, and T_x_ is a surface functional group (–O, –OH, or –F). It is widely used in the fields of catalysis [1,2], energy storage [3,4], and sensing [5,6]. Especially in the field of sensing, MXene, as a conductive material for flexible pressure sensors, has the unique advantages of an adjustable two-dimensional layered structure, high conductivity, good mechanical properties (high flexibility and high strength), hydrophilicity, and a large specific surface area, so it is very competitive [5,7,8].

The flexible pressure sensor based on MXene material shows the characteristics of high sensitivity, wide detection range, and good stability. In addition, as the core component of wearable electronic products, it also needs to have the characteristics of biocompatibility, air permeability, and wearing comfort [9,10]. The interlayer spacing of accordion-like MXene can easily change under the action of external force, resulting in a change in the number and length of conductive paths, resulting in a more intense response of channel resistance (R_b_). Therefore, MXene can improve the relative resistance of the varistor layer, which is beneficial to improving the sensitivity and detection range of the sensor [11]. However, the flexible pressure sensor containing only MXene material is limited to the measurement of small pressure. After exceeding a 0.82% compression range, it quickly reaches saturation, and the interlayer spacing is basically no longer reduced [12]. In most reports, MXene material is attached to the elastic polymer substrate by dip coating or spraying as the pressure-sensitive layer of the flexible pressure sensor [13,14,15]. The conductive modification of the compressible structure of the pressure-sensitive layer is considered to be one of the strategies to improve the performance of the sensor [16]. The current compressible structures include porous skeleton structures [17,18], cotton or fabric [19,20], nonwoven structures [21], and so on. For example, Cheng et al. [22] prepared MXene/PPNs/TPUEM nanofiber mats by the vacuum filtration of poly (styrene-methacrylic acid) @polypyrrole nanospheres (PPNs) and MXene onto a thermoplastic polyurethane electrospun membrane (TPUEM) with a multi-void structure, and assembled a flexible pressure sensor. At the same time, thanks to the conductive network formed by PPNs and MXene on TPU nanofibers, this pressure sensor has a high sensitivity of 16.72 kPa^−1^. However, the compressibility of its single-layer fiber film is limited, resulting in a detectable pressure range for the sensor (~32.1 kPa) that is not wide enough. In addition, the hydrophilic end-capping functional groups (–OH, –F) of MXene make it easy to combine with the polymer substrate, which helps to produce stable contact, reduce the fluctuation of channel resistance (R_b_) and contact resistance (R_c_), and maintain the long-term stability of the sensor [23]. The polymer materials that have been reported as substrates include polyvinyl alcohol (PVA) [24,25], thermoplastic polyurethane (TPU) [26,27], polyvinylidene fluoride (PVDF) [28], etc. However, nano-sized MXene sheets have structural defects and surface-rich functional groups, which are thermodynamically metastable, making them easy to oxidize to more stable metal oxides in the presence of environmental oxygen and water, while their high conductivity and other performance degrades [29]. Therefore, oxidation hurts the long-term stability and effectiveness of the sensor [30]. The oxidation also destroys the lamellar structure and surface functional groups of MXene, reducing the interaction with the polymer substrate.

Researchers were committed to using encapsulation or hydrophobic modification to reduce or even prevent the instability of MXene oxidation [31]. Zhang Z et al. [32] used polydimethylsiloxane (PDMS) to encapsulate the MXene/TPU sensing element. The sensing element was isolated from the external environment using the hydrophobic and dustproof properties of PDMS, which improved the stability of the sensor. A flexible strain sensor constructed by MXene-TPU/PDMS with good durability (>3000 times) was obtained. Nevertheless, the package will inevitably passivate the fractional resistance change in flexible pressure sensors (FPS) and lose the overall sensing performance. Packaging will also reduce air permeability and reduce the comfort of wearing the material. The hydrophobic modification of the MXene surface is a promising method to effectively shield oxygen and water and improve the environmental stability of MXene without compromising its electrical properties [33]. Lee et al. prepared a highly ordered MXene free-standing film by the self-triggered polymerization of polydopamine on the surface of MXene flakes, which has a synergistic synchronization of mechanical and electrical properties and environmental stability. They also demonstrated that polydopamine further promoted the inherent high conductivity and excellent electromagnetic interference shielding performance of MXene membranes [34]. However, hydrophobic modification will cause the loss of hydrophilic groups on the surface, so that MXene loses the ability to interact well with the polymer matrix and cannot adhere firmly to it. This will lead to the stability problem of the pressure sensor in the long-term pressure monitoring process.

Here, a multilayered f-MXene/MXene/TPU (M-fM^2^T) composite with improved environmental stability and pressure sensing performances is successfully fabricated using a hydrogen-bond self-assembly strategy associated with electrostatic attraction. As shown in Figure 1a, the as-proposed M-fM^2^T composite consists of a bottom-layer electrospun TPU fiber mat with a microporous structure, a middle-layer conductive MXene nanosheet which realizes the anchoring and uniform loading on the TPU fibers through hydrogen bonding self-assembly using a high-pressure spraying method, and a top-layer f-MXene nanosheet acting as the anti-oxidized layer, which has strong electrostatic interaction with the middle-layer MXene nanosheet. Further, we cyclically electrospin the bottom-layer TPU fiber mat and successively spray the middle-layer MXene and top-layer f-MXene, stacking them to create a multiple M-fM^2^T composite architecture. Owing to the electrospun microporous TPU fiber and the interlaminar nanoporous structure, the hierarchical micro-nano porous structure of the stacked multiple M-fM^2^T composite architecture can generate continuous changes in the conductive pathways within a wide loading range, ensuring excellent sensitivity and broader pressure detection capability. The pressure sensitivity of our constructed multilayered f-MXene/MXene/TPU piezoresistive sensor (M-fM^2^TPS) can reach 40.31 kPa^−1^ in the range of 0 Pa to 2.4 kPa, and 0.0094 kPa^−1^ (2.4 kPa–120 kPa). Noticeably, the sensitivity and pressure range of our proposed M-fM^2^TPS can be optionally tailored by changing the electrospinning quantity of the spatial layers. Additionally, attributed to the complete porous structure, the air permeation of our M-fM^2^TPS exhibits 0.098 g (average per day), guaranteeing a comfortable and air-permeable wearing property. We also constructed a prototype to explain the potential applications of the as-fabricated M-fM^2^TPS in the field of motion monitoring and medical electronics for conceptual verification.

## 2. Materials and Methods

### 2.1. Materials

Ti_3_AlC_2_ (MAX) was provided by Jilin Yiyi Technology Co., Ltd., Changchun, China; TPU particles (industrial grade) were purchased from BASF Chemical (Shanghai, China) Co., Ltd. Lithium fluoride (LiF, AR, 98%), concentrated hydrochloric acid (HCl, 36–38%), anhydrous ethanol, N, N-dimethylformamide (DMF), and tetrahydrofuran (THF) were provided by Sinopharm Chemical Reagent Co., Ltd. (Shanghai, China); acetic acid (AR, 99.5%) was purchased from Shanghai McLean Biochemical Technology Co., Ltd. (Shanghai, China); and [3-(2-Aminoethylamino)propyl]trimethoxysilane (96%) was supplied by Beijing Enoch Technology Co., Ltd. (Beijing, China). All chemicals and materials were used directly without any treatment.

### 2.2. Preparation of MXene Suspension

The preparation of the Ti_3_C_2_T_X_ MXene nanosheet suspension for high-pressure spraying involved the following: Ti_3_C_2_T_X_ MXene nanosheets were prepared by selective etching and layering methods. A total of 2 g of LiF powder and HCl (40 mL, 9 M) solution were dispersed into a polytetrafluoroethylene (PTFE) container and magnetically stirred for 30 min, allowing LiF to react fully with HCl. The 2 g MAX was slowly added to the mixed solution, and the aluminum layer of the reactant was corroded by magnetic stirring at 35 °C for 24 h. Subsequently, the product after the reaction was centrifuged at 3500 rpm in a centrifuge. After centrifugation, the liquid was poured, and deionized water was added, shaking well. The above steps were repeated until the pH value was greater than or equal to 6. The centrifuged sediments were collected, added to 40 mL deionized water, and sonicated for 1 h under ice water bath conditions. The mixed solution after ultrasonication was centrifuged at 3500 rpm for 30 min to obtain the supernatant, which was the exfoliated MXene suspension. The content of the final Ti_3_C_2_T_X_ MXene nanosheet suspension was determined by vacuum filtration of 1 mL solution, which was about 20 mg mL^−1^.

### 2.3. Preparation of f-MXene Nanosheet Suspension

The preparation of the f-MXene nanosheet suspension for high-pressure spraying involved the following: A total of 5 mL of 20 mg mL^−1^ MXene solution was dispersed in 30 mL of anhydrous ethanol, and 200 mg of [3-(2-Aminoethylamino)propyl]trimethoxysilane was mixed with 15 mL of anhydrous ethanol. In the first 2 h of stirring, the [3-(2-Aminoethylamino)propyl]trimethoxysilane mixed solution was slowly added dropwise to the MXene anhydrous ethanol mixed solution and magnetically stirred for 8 h under the protection of a nitrogen atmosphere. The product after the reaction was centrifuged at 3500 rpm for 30 min, and anhydrous ethanol was added, shaking well. The above steps were repeated to remove the excess silane. The obtained precipitate was dispersed in a 1:1 anhydrous ethanol/aqueous solution. The dispersed product was treated with a probe-type ultrasonic cell disruptor with an energy percentage of 40%, ultrasonic vibration for 3 s, then stopped for 2 s, for a total of 5 min, and the treated solution was colloidal. Finally, the pH value of the mixed solution was adjusted to 3.5 with acetic acid to keep it stable as a colloidal solution. The content of the obtained f-MXene nanosheet suspension was about 2 mg mL^−1^.

### 2.4. Preparation of M-fM^2^TPS

The preparation of the single-layer TPU fiber mat by electrospinning involved the following: TPU particles were first dried at 60 °C for 24 h to remove moisture. The TPU particles were added to the DMF/THF mixed solvent (V_DMF_:V_THF_ = 1:1), and the 10 wt% TPU solution was prepared by magnetic stirring for 6 h. Then, electrospinning was performed with the following parameters: the needle size was 19, the voltage was set to 20 kV, the extrusion speed was 1 mL/h, the distance between the syringe needle and the metal drum receiver was 15 cm, and the rotation speed of the metal drum was 500 r/min. Finally, a 10 cm × 10 cm square TPU fiber mat was obtained.

The preparation of the self-assembled M-fM^2^T composite involved the following: As shown in Figure 1b, after a layer of 10 cm × 10 cm square TPU fiber mats was prepared, an industrial air pump was immediately used on it for high-pressure spraying at a pressure of 0.62 MPa. First, a layer of MXene suspension with a total content of 5 mL MXene was uniformly sprayed on the surface of a 100 cm^2^ square TPU fiber mat. Then a layer of f-MXene suspension with a total content of 0.05 mL/cm^2^ was uniformly sprayed. The above process is defined as cyclic self-assembly. In the process of cyclic self-assembly, external pressure on the fiber mat should be avoided as much as possible to reduce the influence on the performance of the sensor during the assembly process. The M-fM^2^T composite was obtained after 4, 5, 6, and 7 cycles of self-assembly, named fM^2^T4, fM^2^T5, fM^2^T6, and fM^2^T7, respectively. The multilayer composite film was cut into a square of 2 cm × 2 cm, and the copper tape was adhered to the upper and lower surfaces of the multilayer composite film as electrodes. Then a layer of 4 cm × 4 cm square TPU fiber mat was prepared by electrospinning to cover the upper and lower surfaces. The obtained M-fM^2^TPS based on fM^2^T7 had a total mass of 0.488 g and a total thickness of 0.55 mm.

### 2.5. Characterization and Measurement

The crystal structures of MAX, MXene, and f-MXene were analyzed by X-ray powder diffraction (XRD, Empyrean, Malvern Panalytical, Almelo, The Netherlands). MXene and f-MXene were characterized by X-ray photoelectron spectroscopy (XPS, PHI5000 VersaprobeI, ULVAC-PHI, Tokyo, Japan). The microstructure, element content, and distribution of MXene nanosheets and M-fM^2^TPS samples were obtained by scanning electron microscope (SEM, Zeiss GeminiSEM 500, Carl Zeiss AG, Oberkochen, Germany) (SEM, SU1510, Hitachi Co., Tokyo, Japan), focused ion beam electron beam double beam scanning electron microscope (FIB-SEM, Tescan SOLARIS, TESCAN, Prague, The Czech Republic), and its supporting energy dispersive spectrometer (EDS). The chemical structure of the film was characterized by Fourier transform infrared spectrometer FT-IR, NICOLET 5700 FT-IR Spectrometer (Thermo Fisher Scientific, Waltham, America). The samples used for XRD were prepared by vacuum-assisted filtration (VAF). XRD examined the crystal structure of the sample in the range of 5–90° with a step of 5°/min. The MXene sample for SEM images was prepared by dropping 0.05 mg/mL MXene solution on a silicon wafer. The initial MXene concentration was determined by vacuum-assisted filtration (VAF). The resistance change in the sensor under pressure was recorded in real-time using a system composed of a material universal testing machine CMT6103 and a digital bridge VC4092A.

Air permeability test: Vaseline was used to cover different substrates on a bottle containing deionized water (1 g). Then, the experiment was carried out at a temperature of 25 °C and a humidity of 50%. The quality of the remaining water in the bottle was weighed every 24 h to indicate air permeability.

Arterial pulse measurement: Arterial pulse was measured in a sedentary state. To detect wrist radial artery pulsation, the subjects sat for more than 5 min.

Ethical statement: Experiments involving human subjects were performed with the full informed consent of the volunteers. All reported tests meet the ethical requirements of the Hubei University of Technology.

## 3. Results and Discussion

### 3.1. Characterization of MXene and f-MXene

As shown in the SEM image in Figure 2a, the high-quality MXene (Ti_3_C_2_T_X_) nanosheets prepared exhibit a clear sheet structure with a lateral length of 1–2 μm. These MXene nanosheet suspensions exhibit a significant Tyndall effect under the irradiation of a laser beam, as shown in Figure 2b, indicating that the prepared MXene nanosheets can be fully dispersed in water. Moreover, the suspensions of MXene and f-MXene can still exhibit the Tyndall effect after 24 h, revealing that MXene and f-MXene nanosheets can be stably dispersed in water for a long time, which is beneficial to the uniform distribution of high-pressure sprayed coatings.

The results of X-ray diffraction (XRD) are shown in Figure 2c, in which the diffraction peak of the (002) crystal plane of Ti_3_C_2_T_X_ MXene moves from 9.5° to 7.1° of the MAX (002) crystal plane, which is attributed to the intercalation effect. The diffraction peak of about 39° in MAX basically disappeared in MXene, indicating that aluminum was completely etched and layered. These results demonstrate the successful preparation of high-quality MXene nanosheets. The (002) crystal plane diffraction peak of f-MXene is at 7.0°, which is close to the (002) peak of MXene. The reason for the decrease in the f-MXene (002) peak is that the surface functionalization of MXene nanosheets by silanization reagent increases the interlayer distance of MXene nanosheets, and the silanization reaction mainly occurs outside MXene nanosheets [35]. The silanization reaction begins with the hydrolysis of triethoxy to trihydroxyl, and then the co-condensation between the silane and the –OH on the surface of MXene occurs. The [3-(2-Aminoethylamino)propyl]trimethoxysilane will release the methoxy group in the form of methanol as a reaction by-product during the hydrolysis process. A hydrogen bond is formed between the silanol group and the surface-terminated functional group –OH of MXene, and then a covalent bond is formed between MXene and the hydrolyzed [3-(2-Aminoethylamino)propyl]trimethoxysilane [33], as shown in Figure 1b. That is,
(1)$${(\mathrm{CH}}_{3}{O)}_{3}{\mathrm{Si}(\mathrm{CH}}_{2}{)}_{3}{\mathrm{NH}(\mathrm{CH}}_{2}{)}_{2}{\mathrm{NH}}_{2}+{\;H}_{2}O\overset{\;}{\to }{\left(\mathrm{OH}\right)}_{3}{\mathrm{Si}(\mathrm{CH}}_{2}{)}_{3}{\mathrm{NH}(\mathrm{CH}}_{2}{)}_{2}{\mathrm{NH}}_{2}+{\;\mathrm{CH}}_{3}\mathrm{OH}\;$$
(2)$${\left(\mathrm{OH}\right)}_{3}{\mathrm{Si}(\mathrm{CH}}_{2}{)}_{3}{\mathrm{NH}(\mathrm{CH}}_{2}{)}_{2}{\mathrm{NH}}_{2}+{\mathrm{Ti}}_{3}C_{2}{\left(\mathrm{OH}\right)}_{X}\overset{\;}{\to }{\mathrm{Ti}}_{3}C_{2}O_{3}{\mathrm{Si}(\mathrm{CH}}_{2}{)}_{3}{{\mathrm{NH}}_{2}(\mathrm{CH}}_{2}{)}_{2}{\mathrm{NH}}_{3}+{\;H}_{2}O$$

It can be seen from Equation (2) that the silanization reaction consumes the hydrophilic end-capped functional group –OH on the surface of MXene. This will result in f-MXene being unable to interact with TPU fibers through hydrogen bonds like MXene. Accordingly, f-MXene nanosheets are difficult to adhere tightly to the surface of TPU fibers and are easy to peel off when used.

The elemental composition and electronic state of MXene and f-MXene were analyzed by XPS. As shown in Figure 2d, the XPS spectra of the MXene sample can be observed on the F, O, Ti, and C element peak signal, and the new N, Si signal peaks are observed in the f-MXene sample, demonstrating that the surface functionalization of the f-MXene nanosheets is successful. As shown in Figure 2e, 284.74 eV, 282.1 eV, and 286.1 eV in the MXene C 1s spectrum represent the C–C, C–Ti–T_z,_ and CO, respectively. These bond states show the chemical structure of MXene. The f-MXene C 1s spectrum adds a peak of 288.03 eV, corresponding to the COO in Figure 2e; the f-MXene O 1s spectrum adds a characteristic peak, corresponding to the C–O peak in Figure 2f; the spectrum of f-MXene N 1s shows peaks at 399.66 eV and 401.44 eV, corresponding to –NH_2_ and –NH_3_^+^ groups in Figure 2g; the characteristic peak at 102.27 eV on the f-MXene Si 2p spectrum, corresponding to the Si–O–Si in Figure 2h, is the characteristic peak of the functionalized MXene [35]. These bond states show the chemical structure of f-MXene, which is direct evidence of the successful surface functionalization of f-MXene nanosheets.

### 3.2. Characterization of TPU Fiber Mats

In the SEM image of the upper surface of the M-fM^2^T composite (Figure 3a), it can be observed that the fiber network structure is well preserved and the upper surface is uneven. Compared with the smooth surface in Appendix A, the surface of the fibers after the high-pressure spraying of MXene and f-MXene is uniformly and tightly attached to a layer. The EDS spectrum analysis is shown in Figure 3b, where the presence of Ti element indicates the distribution of MXene nanosheets on the fibers, while the presence of Si and N elements indicates the distribution of f-MXene nanosheets. It can be proved that MXene and f-MXene are uniformly attached to the surface of TPU fibers. In addition, it can be found that a layer of MXene and f-MXene is also attached to the pores between the fibers. This can also be seen in the cross-section of the M-fM^2^T composite, as shown in the SEM image of Figure 3c. This shows that the high-pressure spraying suspension process we used can make MXene nanosheets and f-MXene nanosheets penetrate the TPU 3D fiber network. This can further enhance the conductivity of the TPU fiber mats, thereby improving the sensitivity of the device. Since the cross-section was formed by blade cutting, it can be seen that some of the cut-off highly elastic TPU fiber ends are rebounded into clumps. The original thickness of the single-layer f-MXene/MXene/TPU composite is about 20 microns. A high ratio of micron-sized pores and some long cracks with a height of about 10 microns can also be observed in the cross-section. We judge that this is an interlayer pore, which is attributed to the fact that during the layer-by-layer self-assembly of M-fM^2^TPS, the binding force between the adjacent layer, that is, the new layer of electrospun TPU composite and the TPU fiber mats with f-MXene/MXene coating on the surface, became weaker, so larger micropores were introduced between the adjacent layers.

A layer of MXene and a layer of f-MXene suspension were sprayed on the surface of the TPU fiber mat layer by layer, and the samples before and after spraying were analyzed by infrared spectroscopy, as shown in Figure 3d. The typical absorption peaks near 2963 and 3336 cm^−1^ in the TPU fiber mat samples are from the C–H stretching vibration and N–H stretching vibration of the ester. The peak at 1057 cm^−1^ corresponds to the stretching vibration of C–O–C, and the peak near 1731 cm^−1^ belongs to –H–N–COO–. The peak near 1533 cm^−1^ corresponds to the bending vibration of N-H. In contrast, the C=O and N–H peaks of the MXene/TPU composite samples shift to lower wavelengths, 3324 cm^−1^, 1729 cm^−1^, and 1076 cm^−1^, respectively, indicating that the C=O and N–H groups of thermoplastic TPU fibers establish an interaction with the functional groups of Ti_3_C_2_T_X_ MXene nanosheets through hydrogen bonds, that is, MXene nanosheets are loaded onto TPU fibers through hydrogen bond self-assembly [36]. Compared with them, the peaks of f-MXene/MXene/TPU composite move to lower wavelengths again, 3319 cm^−1^, 1727 cm^−1^, and 1061 cm^−1^, respectively, manifesting that interaction between MXene and f-MXene is established by electrostatic attraction [35].

From the above characterization results, we can infer the formation mechanism of the M-fM^2^T composite. As described in Section 3.1 of this paper, the silanization reaction consumes the hydrophilic end-capping functional groups on the surface of MXene, making it difficult for f-MXene nanosheets to adhere closely to the surface of TPU fibers and easy to peel off. To this end, we used the high negative ζ-potential of MXene, abundant hydrophilic end-capping functional groups (–OH, –F) on the surface, and colloidal stability to realize the self-assembly of f-MXene nanosheets on TPU fibers, as shown in Figure 1b. Moreover, the fiber structure prepared by electrospinning technology has a high specific surface area and also provides a uniform and rich adhesion point for MXene. Self-assembly between the abundant hydrophilic groups on the surface of MXene and TPU fibers was carried out by hydrogen bonding, and self-assembly between f-MXene (positively charged protonated amino group of silanes coupling agent on the surface) and MXene (high negative potential on the surface) was carried out by electrostatic attraction (Figure 1b). Through the layer-by-layer self-assembly strategy, strong interactions were established between f-MXene nanosheets, MXene nanosheets, and TPU fibers by hydrogen bonding or electrostatic attraction, which effectively achieved strong adhesion and uniform loading between f-MXene, MXene, and TPU fibers.

### 3.3. Sensing Performance

The sensing performance indexes of FPS mainly include sensitivity, pressure sensing range, response/recovery time, stability, and long-term reliability. In addition, to be applied to wearable electronic products, FPS also needs to have good biocompatibility, skin conformal ability, portability, breathability, and other characteristics [37]. Among them, the key parameters are sensitivity and pressure detection range. Here we define the fractional resistance change value of the sensor as the percentage of the resistance change after the load pressure on the sensor surface to the initial resistance (ΔR/R_0_, %), ΔR = |R − R_0_|, where R_0_ is the initial resistance, and ΔR is the resistance change after the load pressure on the sensor surface. Sensitivity is a key parameter for evaluating the performance of pressure sensors. The sensitivity of the sensor is defined as the slope of the resistance change and the pressure change S = δ (ΔR/R_0_)/ΔP, where ΔP is the pressure load change on the sensor surface.

We prepared M-fM^2^TPS based on fM^2^T4, fM^2^T5, fM^2^T6, and fM^2^T7, and plotted the relationship curve (ΔR/R_0_-ΔP) between the surface load pressure and fractional resistance change value of these sensors to determine the effect of the number of layers on the sensitivity and sensing range of the sensor (Figure 4a). Comparing the curves in the figure, it can be found that the sensitivity of M-fM^2^TPS of fM^2^T4 is as high as 40.31 kPa^−1^ in the low-pressure range (0 Pa to 2.4 kPa), which is higher than that of fM^2^T5, fM^2^T6, and fM^2^T7 (28.42 kPa^−1^, 6.9 kPa^−1^, and 10.04 kPa^−1^, respectively). The maximum pressure detection limit (120 kPa) of the M-fM^2^TPS of fM^2^T7 is much higher than that of fM^2^T4, fM^2^T5, and fM^2^T6 (2.4 kPa, 3.6 kPa, and 60 kPa, respectively). Therefore, by adjusting the number of layers of the f-MXene/MXene/TPU composite prepared by cyclic self-assembly, a high sensitivity of 40.31 kPa^−1^ can be obtained in a lower pressure range (0 Pa to 20 kPa); a wide pressure sensing range (0 Pa to 120 kPa) can also be obtained.

To explore the minimum pressure detection limit, response, and recovery time of M-fM^2^TPS response time, TPU particles (20 Pa) were placed on M-fM^2^TPS, left for about 3 s, and then removed to obtain the fractional resistance change curve in Figure 4b. The curve shows a significant step, and the response signal and noise signal can be identified, indicating that the minimum pressure detection limit of M-fM^2^TPS may be 20 Pa, which can detect slight external pressure. The curve also shows that the transient response time of M-fM^2^TPS is 180 ms and the recovery time is 200 ms. To test the stability of M-fM^2^TPS, a cyclic pressure of 1~100 kPa and a frequency of 2 mm/min was applied on the surface of M-fM^2^TPS. The fractional resistance change curve in Figure 4c shows that M-fM^2^TPS can achieve a stable response under dynamic external pressure stimulation at different frequencies. For the benefit of testing the long-term reliability of the sensor performance, a 50 kPa pressure loading–unloading cycle was performed on M-fM^2^TPS for 1000 cycles. According to the fractional resistance change output curve in Figure 4e, the fractional resistance change signal under long-term pressure load is stable and repeatable, showing its good robustness.

The resistance of M-fM^2^TPS consists of three parts: the channel resistance (R_b_) in the pressure-sensitive layer (M-fM^2^T composite), the contact resistance (R_c_) between it and the copper tape electrode, and the intrinsic resistance (R_e_) of the copper tape electrode. Among them, the copper tape electrode has high conductivity and small resistance (R_eXY_ = 0.2 Ω/sq, R_eZ_ = 0.05–0.08 Ω/sq), and its pressure-sensitive characteristics are negligible, so fractional resistance change in M-fM^2^TPS comes from R_b_ and R_c_. It can be observed from the SEM image of the upper surface of the M-fM^2^T composite (Figure 3a) that the fibers are interwoven to form a convex and concave surface. There are only a few contact points between this surface and the copper tape electrode to form a conductive path. The initial channel resistance R_c0_ in the M-fM^2^T4 composite is 400 kΩ, which has a higher basic resistance value than the 200 Ω of R_c0_ in the single-layer fiber mat, resulting in a higher signal-to-noise ratio for M-fM^2^TPS. Consequently, the initial resistance R_b0_ and R_c0_ of M-fM^2^TPS are high. As the surface load of M-fM^2^TPS gradually increased the contact area between the f-MXene/MXene on the surface of the pressure-sensitive layer and the copper tape electrode also increased, there was a resulting decrease in R_c_. When the surface load increased to a certain amount, R_c_ no longer decreased. Furthermore, the M-fM^2^T composite self-assembled layer by layer by electrospinning and spraying technology has a three-dimensional multi-level micro-nano pore structure, that is, micro-pores between adjacent layers, micro-pores inside each layer, nano-pores between stacked f-MXene/MXene nanosheets, and nano-gaps between each MXene sheet layer. As the pressure load on the surface of M-fM^2^TPS gradually increased, the compressive deformation of the M-fM^2^T composite pressure-sensitive layer increased. These pores were continuously squeezed and eliminated, and the pore walls were contacted, resulting in the continuous change in the number and length of the conductive paths, leading to a continuous and stable response of R_b_ within a large pressure load range, which broadened the pressure detection range of M-fM^2^TPS. The sensitivity of M-fM^2^TPS is high in the low-pressure range (0 Pa to 20 kPa), which is attributed to the extrusion of a large number of micropores between adjacent layers and within each layer at this stage, resulting in more conductive pathways, so R_b_ decreases significantly. When the external pressure exceeded 20 kPa, most of the micropores had been squeezed and eliminated, and the multilayer composite was further compressed. At this time, the nanogaps between the stacked f-MXene/MXene nanosheets and between each MXenes sheet were mainly squeezed and eliminated. The conductive path gradually tended to be saturated, so R_b_ decreased slowly, and the sensitivity of M-fM^2^TPS was low. In addition, the rich nano-gaps in the interface between the adjacent fabric layers of the multilayer sensor are also beneficial to broaden the pressure detection range. Therefore, the pressure detection range of the pressure sensor also increases with the increase in the number of layers of the sensing material, which indicates the customizability of our pressure sensor. When the external pressure was removed, the highly elastic TPU fiber mat quickly rebounded back to its initial state, and the resistance also recovered. This gives M-fM^2^TPS better cycle durability and reliability.

In addition, through the layer-by-layer self-assembly strategy, strong interactions between f-MXene nanosheets, MXene nanosheets, and TPU fibers were established by hydrogen bonding or electrostatic attraction, which realized the strong adhesion and uniform loading of f-MXene nanosheets and MXene nanosheets on TPU fibers, thereby improving the durability of M-fM^2^TPS during use. In addition, a strong interaction between f-MXene nanosheets and MXene nanosheets was established by electrostatic attraction, which changes the tunnel distance between adjacent MXene and f-MXene. Consequently, a richer effective conductive path was constructed, which further improves the sensing sensitivity.

### 3.4. Environmental Stability and Air Permeability

After preparing a layer of 10 cm × 10 cm square TPU fiber mat, a layer of a MXene suspension of 0.05 mL/cm^2^ was uniformly sprayed on it immediately, and then a layer of suspension was uniformly sprayed. Keeping the total content of the sprayed nanosheets on the surface of the TPU fiber mat unchanged, only the content of the second sprayed f-MXene was changed, and a coating TPU film with the content ratio of MXene to f-MXene of 2:0, 1.5:0.5, 1:1 was obtained. In the coating with a content ratio of 2:0, the second spraying was a 0.05 mL/cm^2^ MXene suspension. The M-fM^2^TPS with the above three content ratios was placed in the natural environment for 30 days, and the resistance naturally placed on the desktop was tested and recorded once a day at the same time. The percentage of the difference between the resistance of the day and the initial resistance measured on the first day to the initial resistance (ΔR/R_0_, %) with the long cycle time was obtained, as shown in Figure 5a. R_0_ is the initial resistance measured on the first day, and ΔR is the difference between the resistance of the day and the initial resistance, ΔR = R − R_0_. It was found that the resistance of the M-fM^2^TPS (completely free of f-MXene) with a content ratio of MXene to f-MXene of 2:0 increases rapidly over time, and the resistance change rate of 30 days is as high as 19.8%. The degradation of the M-fM^2^TPS with a content ratio of 1.5:0.5 slowed down, and the resistance change rate of 30 days also reached 4.6%. The M-fM^2^TPS with a content ratio of 1:1 was the most stable, and the resistance change rate of 30 days was only about 0.8%. This shows that with the increase in f-MXene content, the oxidation resistance of M-fM^2^TPS gradually becomes stronger. This is attributed to the surface functionalization of MXene nanosheets with the hydrophobic silanization reagent [3-(2-Aminoethylamino)propyl]trimethoxysilane. The uniform and dense hydrophobic protective layer formed by the covalent effect of the silanization reaction can effectively prevent water or dissolved oxygen from directly contacting with MXene, making it more conducive to the long-term stability of the sensing performance of M-fM^2^TPS.

To verify the air permeability of M-fM^2^TPS, four control experiments with four bottles containing 1.0 g deionized water were set up. In these experiments, the bottle mouths were uncovered; the cap, PDMS square sheet (2 cm × 2 cm × 0.5 cm), and M-fM^2^TPS were covered; and the water content changes were recorded daily. The results of Figure 5b show that the water content of the sample covered with the bottle cap and PDMS is almost unchanged. The water content of the sample covered with M-fM^2^TPS at the mouth of the bottle changed by 60%, which indicated that the naturally rich three-dimensional multi-level micro-nano pore structure of the fiber mat prepared by electrospinning technology endowed M-fM^2^TPS with excellent air permeability.

### 3.5. Application of M-fM^2^TPS

We designed several experiments to verify the application of M-fM^2^TPS in the real-time monitoring of microdeformation and various human activities. To show the performance of M-fM^2^TPS in small pressure monitoring, a small weight of 1 g is placed on the element. As shown in Figure 6a, we repeatedly placed the small weight on the piezoresistive sensor and then recorded its resistance change in the process. The results show that M-fM^2^TPS is sensitive to light objects. Figure 6b shows the monitoring of finger movement by M-fM^2^TPS. When the finger does a bending motion, the change in ΔR/R_0_ waveform is obtained. At the same time, the bending motion of the knee can also be monitored in real-time (Figure 6c). In addition to detecting the movements of the human finger and knee, we also placed an M-fM^2^TPS on the neck to monitor changes in the resistance of volunteers when coughing and making sounds. As shown in Figure 6d, M-fM^2^TPS monitored vocal cord muscle movement during coughing (left curve) and changes in neck pressure during repeated vocalizations (right curve). Figure 6e reveals the monitoring of human physiological signals. M-fM^2^TPS was tightly attached to the volunteer’s wrist, and the regular waveform generated by the pulse beat could be detected. The enlarged waveform shows three characteristic peaks, corresponding to shock wave (P), tidal wave (T), and diastolic wave (D). These are important indicators for evaluating heart-related diseases [48]. These findings demonstrate the potential of our sensors in non-invasive health monitoring, wearable human-machine monitoring, and exercise training rehabilitation.

## 4. Conclusions

In summary, silane agent was used to surface functionalize MXene to obtain f-MXene. MXene was self-assembled by hydrogen bonding on TPU fiber, and f-MXene was electrostatically self-assembled with MXene. A flexible piezoresistive sensor M-fM^2^TPS with enhanced environmental stability and adjustable sensing performance was successfully prepared. The advantages of this preparation strategy are as follows: (1) The problem of F-MXene nanosheets being easy to peel off was solved. Taking advantage of the high negative ζ-potential of MXene, abundant hydrophilic end-capping functional groups (–OH, –F) on the surface, and colloidal stability, strong interactions between f-MXene, MXene, and TPU fibers were established by hydrogen bonding or electrostatic attraction, which realized the strong adhesion and uniform loading between f-MXene nanosheets and TPU fibers, thereby improving the durability of M-fM^2^TPS during use. (2) The problem of the invalid stacking of MXene was overcome. Electrostatic self-assembly changes the tunnel distance between adjacent MXene and f-MXene, thus constructing a richer effective conductive path and further improving the sensing sensitivity. (3) The problem of the easy oxidation of MXene was overcome. The uniform and dense hydrophobic protective layer formed by hydrophobic silanization can effectively prevent the direct contact of water or dissolved oxygen with MXene. Thus, it is more conducive to the long-term stability of the sensing performance of M-fM^2^TPS. (4) This inspired the construction of a three-dimensional multi-level micro-nano pore structure. Electrospinning technology is conducive to the formation of a three-dimensional multi-level micro-nano pore structure using an M-fM^2^T composite, that is, micro-pores between adjacent layers, micro-pores inside each layer, nano-pores between stacked f-MXene/MXene nanosheets, and nano-gaps between each MXene sheet. These pores can generate continuous changes in the number and length of conductive paths within a large pressure load range to obtain continuous channel resistance response changes within a large pressure range, broadening the pressure detection range of M-fM^2^TPS. Moreover, the multi-layer TPU fiber mat prepared by electrospinning technology is used as a flexible matrix, which causes M-fM^2^TPS to have good flexibility, air permeability, biocompatibility, and satisfactory wearing comfort. By adjusting the number of layers of M-fM^2^TPS prepared by cyclic self-assembly, a high sensitivity of 40.31 kPa^−1^ can be obtained in a lower pressure range (0 Pa to 20 kPa); a wide pressure sensing range (0 Pa to 120 kPa) can also be obtained. It exhibits good environmental stability and long-term robustness, and its resistance fluctuation is only about 0.8% after being placed in the natural environment for 30 days, and no signal damping phenomenon occurs under a pressure loading–unloading of more than 1000 cycles. Its advantages also include softness, lightness, and breathability. Consequently, FPS shows broad application prospects in the fields of intelligent motion monitoring and medical electronics.

## Figures and Tables

**Figure 1 polymers-16-01337-f001:**
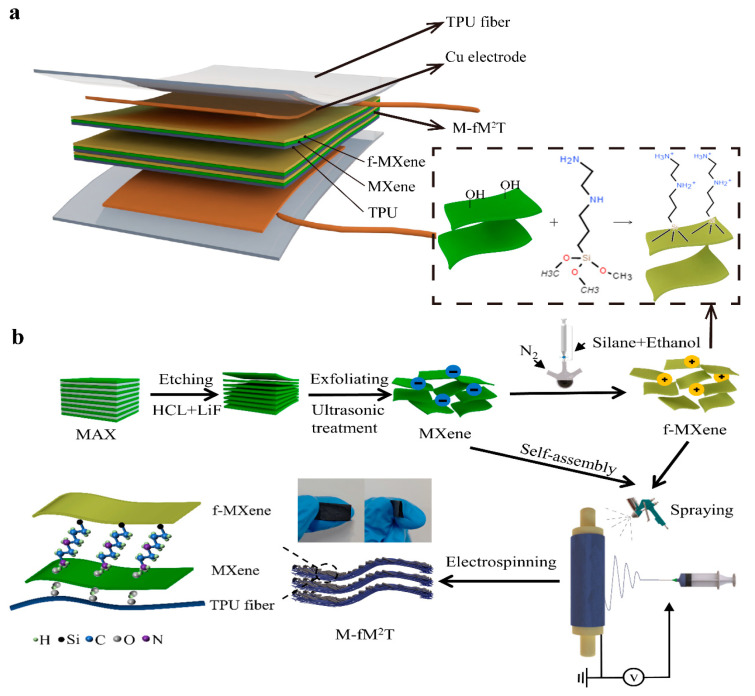
Schematic illustration for the construction and synthesis of the M-fM^2^TPS and the M-fM^2^T composite. (**a**) Construction of the M-fM^2^TPS. (**b**) Synthesis of the M-fM^2^T composite, and the corresponding hydrogen bonding and electrostatic self-assembly strategies.

**Figure 2 polymers-16-01337-f002:**
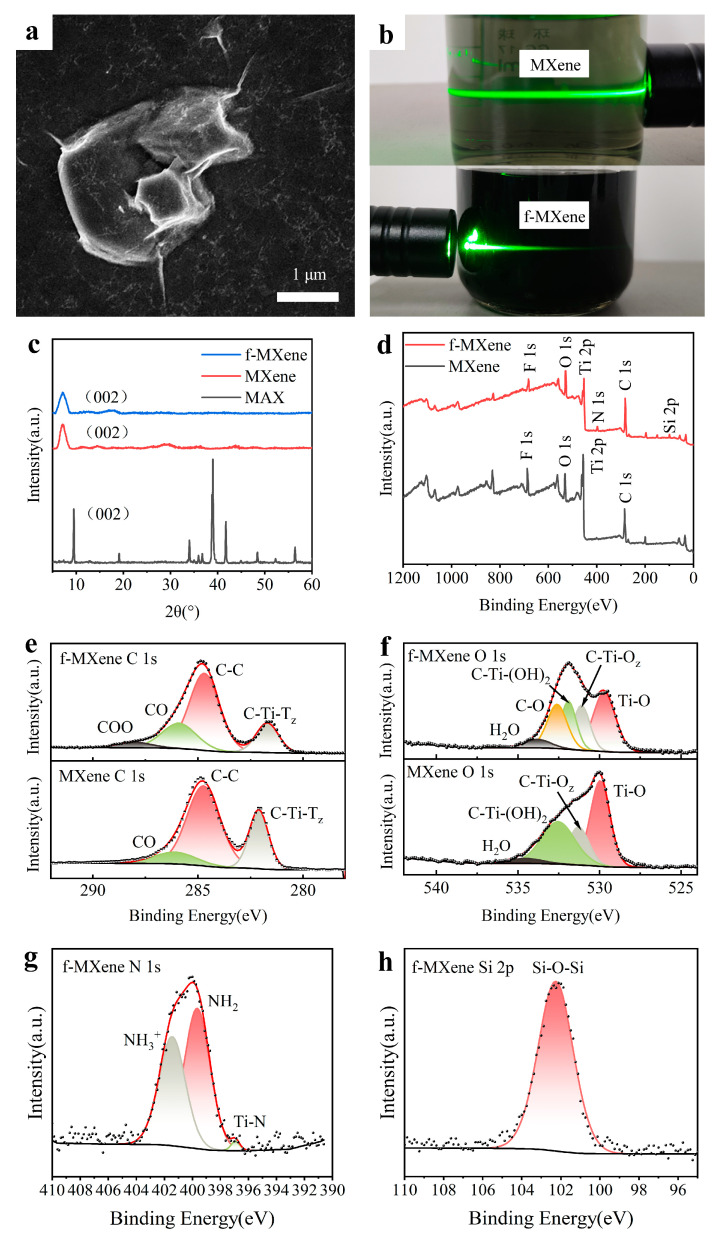
Characterization of MXene and f-MXene. (**a**) SEM image of prepared MXene (Ti_3_C_2_T_X_) nanosheets. (**b**) Photo of MXene and f-MXene solutions used for high-pressure spraying illuminated by a laser beam. (**c**) XRD characterization of MAX, MXene, and f-MXene. (**d**–**h**) XPS analysis of MXene and f-MXene.

**Figure 3 polymers-16-01337-f003:**
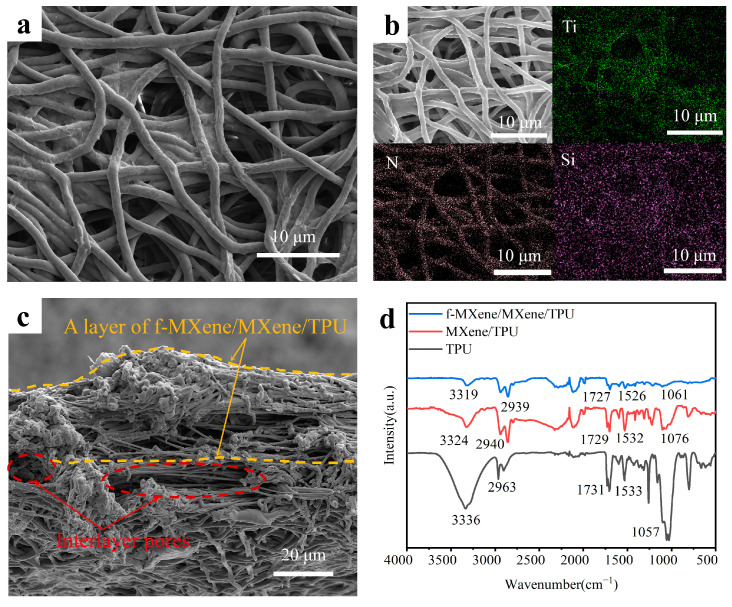
Microstructure and interface interaction of the M-fM^2^T composite. (**a**) SEM image of the surface of the M-fM^2^T composite. (**b**) EDS spectra of an M-fM^2^T composite fiber mat. (**c**) SEM image of the cross-section of the M-fM^2^T composite. (**d**) FT-IR spectra of TPU, MXene/TPU, and f-MXene/MXene/TPU composites.

**Figure 4 polymers-16-01337-f004:**
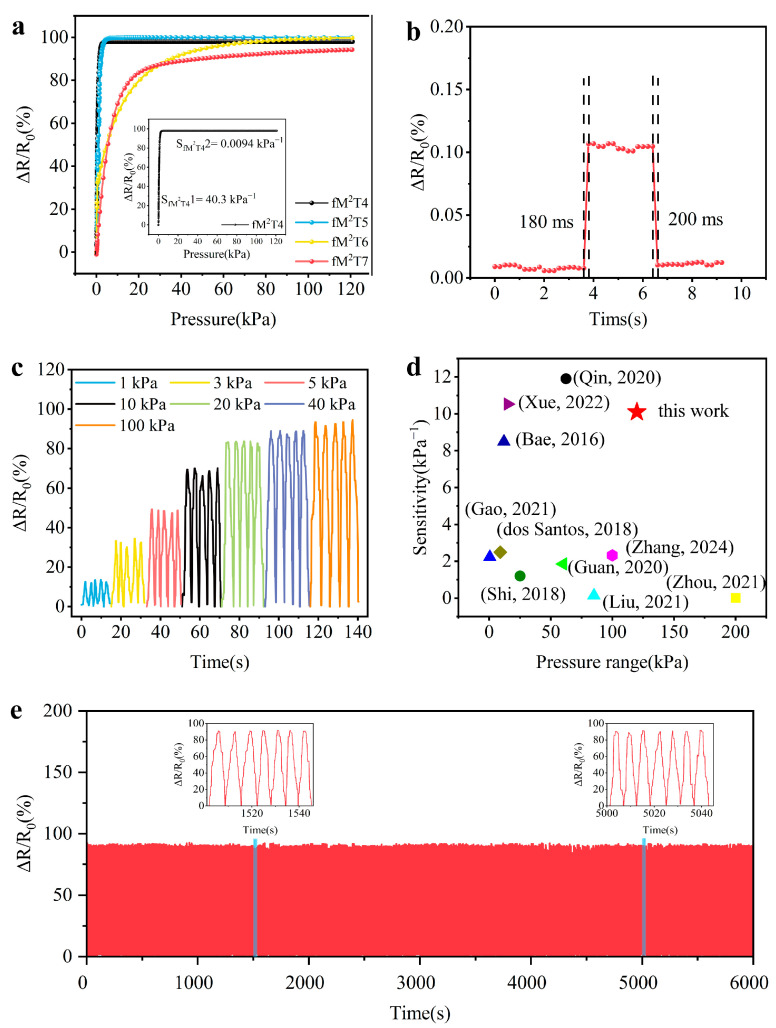
Pressure sensing performance of M-fM^2^TPS. (**a**) Fractional resistance change curve (ΔR/R_0_-ΔP) and sensitivity (δ (ΔR/R_0_)/ΔP) of M-fM^2^TPS based on fM^2^T4, fM^2^T5, fM^2^T6, and fM^2^T7. (**b**) Response and recovery time of M-fM^2^TPS under 20 Pa pressure load. (**c**) Fractional resistance change curve of M-fM^2^TPS under 1~100 kPa pressure loading and cyclic compression at a frequency of 2 mm/min. (**d**) Comparison of the sensing range and sensitivity of M-fM^2^TPS with pressure sensors reported in previous years [38,39,40,41,42,43,44,45,46,47]. (**e**) Fractional resistance changes of M-fM^2^TPS during 1000 repeated loading–unloading cycles at 50 kPa pressure.

**Figure 5 polymers-16-01337-f005:**
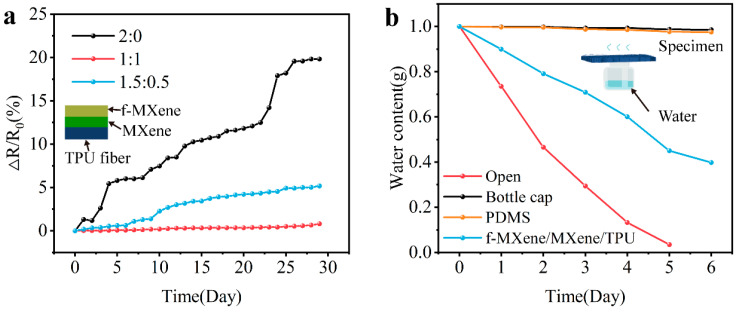
Oxidation resistance and air permeability of M-fM2TPS. M-fM^2^TPS of coatings with different content ratios of f-MXene and MXene. (**a**) Variation curve of the difference between the resistance of the day and the initial resistance measured on the first day as a percentage of the initial resistance (ΔR/R_0_, %) with time. (**b**) Change curve of permeability of M-fM^2^TPS with time.

**Figure 6 polymers-16-01337-f006:**
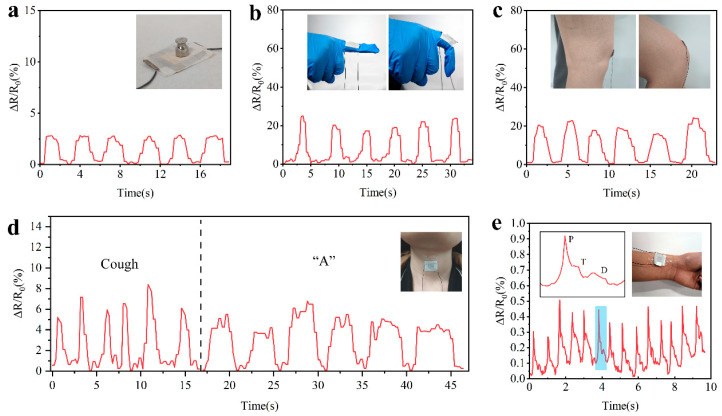
Application demonstration of M-fM^2^TPS in different scenarios. M-fM^2^TPS real-time detection of resistance response signals of (**a**) light objects, (**b**) finger bending, (**c**) knee bending, (**d**) cough and pronunciation of “A”, (**e**) radial artery pulse, the blue area is a characteristicl peak of the radial artery pulse, and local magnification view of the characteristic peak.

## Data Availability

Data are contained within the article.

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
