# Peer review of "Anti-Oxidized Self-Assembly of Multilayered F-Mene/MXene/TPU Composite with Improved Environmental Stability and Pressure Sensing Performances"

_polymers, 2024, doi:10.3390/polym16101337_

Round 1

Reviewer 1 Report

Comments and Suggestions for Authors

The paper titled “Anti-oxidized self-assembly of multilayered f-Mene/MXene /TPU composite with improved environmental stability and pressure sensing performances” showcases a multi-layered MXene composite film, firmly attached to TPU fibers. The initial layer of MXene employs hydrogen-bonded self-assembly, followed by the adhesion facilitated by anti-oxidized functionalized MXene through electrostatic attractions. This composite assembly exhibits outstanding sensitivity at 40.31 kPa-1 and extends the sensing pressure range, while also enhancing environmental stability. While the work is conducted well, there are a few areas where clarification is needed for improved understanding.

1.       The experimental section lacks detailed information on the synthetic procedure for MXene functionalization. The authors are urged to include these specifics in Section 2.

2.       The authors state that "Moreover, the suspensions of MXene and f-MXene can still exhibit the Tyndall effect after 24 h, revealing that MXene and f-MXene nanosheets can be stably dispersed in water for a long time, which is beneficial to the uniform distribution of high-pressure sprayed coatings."  However, the corresponding figure for f-MXene is not included.

3.       At the outset of Section 3.2, the authors write “The elemental composition and electronic state of MXene and f-MXene were …….. demonstrating that the surface fu”". This information seems redundant, as it has been previously addressed in line 253.

4.       How did the authors determine the fiber diameter, including the procedures followed, any software utilized (if applicable), and details regarding thresholding options (if employed)?

5.       In line 527, there is a typo error where "highsensitivity" should be corrected to "high sensitivity".

Reviewer 2 Report

Comments and Suggestions for Authors

In this paper, a multilayered f-MXene/MXene/TPU (M-fM2T) composite with improved environmental stability and pressure sensing performances is fabricated using a hydrogen-bond self-assembly strategy, associated with electrostatic attraction. The bottom-layer TPU fiber mat is cyclically electrospined; the middle-layer MXene sprayed and top-layer f-MXene stack a multiple M-fM2T composite architecture. Owning to the electrospun microporous TPU fiber and the interlaminar nanoporous structure, the hierarchical micro-nano porous structure of the stacked multiple M-fM2T composite architecture can generate continuous changes in the conductive pathways within a wide loading range, ensuring sensitivity and broader pressure detection capability. The pressure sensitivity of the constructed multilayered f-MXene/MXene/TPU piezoresistive sensor (M-fM2TPS) can reach 40.31 kPa-1 in the range of 0 – 2.4 kPa, and 0.0094 kPa-1 (2.4 – 120 kPa). The sensitivity and pressure range of the proposed M-fM2TPS can be optionally tailored by changing the electrospinning quantity of the spatial layers. Moreover, a prototype is constructed to explain the potential applications of the as-fabricated M-fM2TPS in the field of motion monitoring and medical electronics.

Some shortcoming and missing of the paper are the following:

1. All abbreviations must be deciphered at first appearance in text and may be presented after main text of the paper.

2. The paper text is located non-optimally in pages 6 and 10.

3. Fig. 3: (b) and (c) must change places with corresponding change of reference text on these figures into the paper.

4. (Line 275): “…demonstrating that the surface fu” – It is noncompleted phrase.

5. (Line 276): “Figure S1 (supporting information, S1)…” – Figure S1 must be presented into text.
